# Constructing the Sulfur-Doped CdO@In_2_O_3_ Nanofibers Ternary Heterojunction for Efficient Photocatalytic Hydrogen Production

**DOI:** 10.3390/nano13030401

**Published:** 2023-01-18

**Authors:** Haiyan Zhang, Zi Zhu, Min Yang, Youji Li, Xiao Lin, Ming Li, Senpei Tang, Yuan Teng, Dai-Bin Kuang

**Affiliations:** 1National Experimental Teaching Demonstration Center for Chemistry, College of Chemistry and Chemical Engineering, Jishou University, Jishou 416000, China; 2MOE Key Laboratory of Bioinorganic and Synthetic Chemistry, Lehn Institute of Functional Materials, School of Chemistry, Sun Yat-sen University, Guangzhou 510006, China

**Keywords:** S-doped, CdO@In_2_O_3_ nanofiber, heterojunction, in-situ electrospinning, photocatalytic hydrogen production

## Abstract

An S-doped CdO@In_2_O_3_ nanofiber was successfully designed by in-situ electrospinning along and subsequent calcination treatment. Under artificial sunlight illumination, the S/CdO@In_2_O_3_-25 displayed a superior photocatalytic hydrogen evolution rate of 4564.58 μmol·g^−1^·h^−1^, with approximately 22.0 and 1261.0-fold of those shown by the S/CdO and S/In_2_O_3_ samples, respectively. The experimental and theoretical analyses illustrate that the unique one-dimensional (1D) nanofiber morphology and rich oxygen vacancies optimized the electronic structure of the nanofibers and adsorption/desorption behaviors of reaction intermediates, contributing to the realization of the remarkable solar-to-H_2_ conversion efficiencies. Moreover, the staggered band structure and intimate contact heterointerfaces facilitate the formation of a type-II double charge-transfer pathway, promoting the spatial separation of photoexcited charge carriers. These results could inform the design of other advanced catalyst materials for photocatalytic reactions.

## 1. Introduction

The consumption of non-renewable energy has aggravated energy shortages and environmental pollution, largely hindering the progress of human civilization. It has been reported that the sustainable and renewable energy sources are key to relieving these issues [1,2]. Among these renewable energy sources, hydrogen has been considered an important energy carrier because of its pollution-free and high energy density characteristics; moreover, it is also expected to act as the most promising alternative fuel in the future [3,4]. Compared with other hydrogen generation technologies, photocatalytic water splitting induced by catalysts to generate hydrogen in large quantities has been considered as one of the most promising strategies to overcome the global energy crisis [5]. Therefore, the solar-to-hydrogen technique has attracted considerable attention. Currently, in order to attain preferable hydrogen evolution activity, the development of efficient and eco-friendly hydrogen photocatalyst materials, which is a challenging task, is necessary [6]. In recent decades, various oxide semiconductor materials for photocatalysis such as In_2_O_3_ [7], ZnO [8], TiO_2_ [9], CdO [10], and CdS [11], have been widely explored. As a typical semiconductor, low-cost CdO is an n-II-VI material with outstanding electrical conductivity and a direct band gap [12,13]. It has been widely used in solar cells [14], transparent electrodes [15], electrochemical capacitors, gas sensors, and as a photocatalyst [16]. In addition, In_2_O_3_ is an important n-type semiconductor widely utilized in optoelectronic devices and photocatalysis because of its suitable band alignment, good photothermal stability, low toxicity, and unique optical/electronic properties [17,18]. Unfortunately, these single photocatalyst materials suffer from rapid recombination of charge carriers and low light utilization rates, which severely hinders their further application [19]. It is well known that the performance of photocatalysts can be improved by morphological adjustment [20], functional interface engineering [21], and heterojunction construction [22], which can greatly overcome the above shortcomings [23,24]. Thus, various advanced heterostructure photocatalysts such as CdTe-Bi_2_S_3_ [25], MoS_2_@CoMoS_4_ [26], Bi_2_S_3_@CoO [27], BiOI/Ag/PANI [28], *α*-Fe_2_O_3_/CeO_2_ [29], ZnO/ACN/MnO_2_ [30], α-Fe_2_O_3_/g-C_3_N_4_ [31], and Bi_2_WO_6_/TiO_2_ [32] have been extensively developed.

In addition to the above-improved strategies, the doping of non-metallic atoms (such as N, C, and S), which could result in the formation of abundant vacancies that could serve as potential charge capture centers that promote the spatial separation of carriers, has also been effective [33,34]. Among the reported micro/nanomorphologies, one-dimensional (1D) nanomaterials exhibit attractive application merits [35]. In particular, nanofibers have the sizeable 1D morphology, highly aligned nanoparticles and a large pore structure, which could provide rich charge-transfer channels for effective photocatalytic reactions. Electrospinning has been proposed to construct various 1D nanofibers materials for photocatalytic hydrogen evolution and dye degradation [36,37,38,39,40].

In this paper, we report the synthesis of an ingenious S-doped CdO@In_2_O_3_ hybrid nanofiber via in-situ electrospinning and facile calcination. As the S-doped CdO@In_2_O_3_ nanofiber was used as a photocatalyst, clean water was effectively reduced to H_2_ within a single integrated system under the simulated sunlight irradiation. The superior photocatalytic hydrogen evolution activity in the S-doped CdO@In_2_O_3_ hybrid can be attributed to its improved sunlight absorption ability, suppressed photo-induced carrier recombination, and accelerated charge separation properties. As an encouraging result, the optimized hydrogen evolution rate of the as-obtained hybrid can reach 4564.58 μmol·g^−1^·h^−1^, which exceeds that of the individual ones. To the best of our knowledge, this is the first study to report the synthesis of an S-doped CdO@In_2_O_3_ ternary heterojunction for efficient H_2_ evolution under simulated sunlight irradiation. The results herein are expected to be of interest to researchers.

## 2. Materials and Methods

### 2.1. Materials and Reagents

Polyvinylpyrrolidone (PVP), N, N-dimethylformamide (DMF), and absolute ethanol (C_2_H_5_OH) were purchased from Beijing Huahengwei Technology Ltd (Beijing, China). Indium nitrate 4.5 hydrate (In(NO_3_)_3_·4.5H_2_O), cadmium nitrate tetrahydrate (Cd(NO_3_)_2_·4H_2_O), and sulfourea were purchased from SINOPHARM (Beijing, China). All chemical reagents were of analytical grade and did not require further purification.

### 2.2. Preparation of Samples

The electrospinning process is illustrated in Figure 1. Specifically, a certain amount of In(NO_3_)_3_·4.5H_2_O, Cd(NO_3_)_2_·4H_2_O and sulfourea with different dosages were dissolved in 2 mL DMF containing 0.8 g PVP and 8 mL ethanol. After stirring for 12 h, a mixed gel containing sulfourea, In(NO_3_)_3_·4.5H_2_O, Cd(NO_3_)_2_·4H_2_O, and PVP was obtained. The prepared precursor sol was then poured into the injector using metal needles. The actual distance between the tip of the needle and Al foil was approximately 17 cm. When a positive voltage of 1.5 kV and a negative pressure were applied, the jet was stretched by an electrostatic force to produce hybrid nanofibers. The as-prepared product was then dried in an oven for 12 h and calcined in a muffle furnace at 480 °C for 4 h. The as-obtained samples were denoted as S/CdO@In_2_O_3_, where the S/CdO@In_2_O_3_ samples with molar ratios of 0.15, 0.25, or 0.35 of In_2_O_3_ and CdO were denoted as S/CI-15, S/CI-25, and S/CI-35, respectively. For comparison, pure In_2_O_3_, CdO, CdO@In_2_O_3_-25 (CI-25), S/CdO (S/C), and S/In_2_O_3_ (S/I) were also synthesized using the same procedure.

### 2.3. Characterization of the Materials

X-ray photoelectron spectroscopy (XPS) (ESCALAB Xi+, Thermo Fisher Scientific, Waltham, MA, USA) was used to estimate the elemental surface chemical states. Specifically, the vacuum degree of the analysis chamber is 8 × l0^−10^ Pa. The excitation source is Al Kα X-ray (hv = 1486.6 eV) with the working voltage of 12.5 kV and the filament current of 16 mA. The results are the signal accumulation of 10 cycles test. The pass energy of the survey spectrum and narrow spectrum is 100 and 30 eV, respectively, with the step size of 0.1 eV and the dwell time of 40–50 ms. The light spot diameter is about 650 μm. All the XPS spectra were calibrated by the 284.8 eV of C1s. Powder X-ray diffraction (XRD) patterns were measured in the 2θ region of 5–80° using a SmartLab SE (Rigaku, Tokyo, Japan) diffractometer with a copper target as the radiation source. Photoluminescence (PL) emission spectra were measured using a F-7000 fluorescence spectrometer (Hitachi, Tokyo, Japan). Fourier transform infrared (FT-IR) spectroscopy was performed using a IS10 (Nicolet, WI, USA) spectrometer. The morphological characteristics of the samples were observed using a Sigma 300 field-emission scanning electron microscope (FE-SEM) (Zeiss, Oberkochen, Germany) (FE-SEM). Transmission electron microscopy (TEM)/high resolution transmission electron microscopy (HR-TEM) was performed using an a Tecnai F20 microscope (FEI, Portland, OR, USA). The solid ultraviolet-visible (UV-vis) diffuse reflectance (UV-vis DRS) spectra were determined using a UV-2600 spectrometer (Shimadzu, Tokyo, Japan). Raman spectroscopy was performed on a Raman spectrometer (Renishaw, London, UK).

### 2.4. The Preparation of Catalyst Films and Photoelectrochemical Measurements

A three-electrode configuration with a quartz battery was used to evaluate the photoelectrochemical properties of the samples. A 0.1 mol/L Na_2_SO_4_ solution was used as the electrolyte, and the prepared catalyst films, platinum wire, and saturated calomel electrode were used as the working, counter, and reference electrodes, respectively. An electrochemical workstation (CHI650E) (Chenhua, Shanghai, China) was used. Specifically, the FTO glass was cleaned ultrasonically in acetone half a hour, rinsed with distilled water and ethanol, and dried at 60 °C. And 10.0 mg catalysts were dispersed in l mL absolute ethanol with 15 μL Nafion solution, and then ultrasound treated for 10 min. The suspension was evenly loading onto the FTO, glass, and then dried at 60 °C under vacuum conditions. The transient photocurrent response test was performed using a 400 W xenon lamp, and the stable time was 20 s for signal acquisition. The electrochemical impedance spectroscopy (EIS) (Chenhua, Shanghai, China) test frequency range was 100 KHz–0.01 Hz, and the initial voltage was measured for the open circuit voltage acquisition signal. The Mott–Schottky curves were tested at 500 and 1500 Hz with the measured open-circuit voltage at the setting center −(0.5–1) V as the starting voltage, +(0.5–1) V as the termination voltage, and 0.01 V as the amplitude.

### 2.5. Photocatalytic Performance Measurement

Photocatalytic hydrogen production as performed in a cylindrically irradiated quartz vessel. Visible light (λ > 400 nm, 300 W Xe, CEL-HXF300) (Zhongjiao Jinyuan Technology Ltd, Beijing, China) was the light source for photocatalytic reactions. The light source was placed at a distance of 12 cm on the surface of the reaction solution, and all UV light with a wavelength less than 400 nm was removed using a 400 nm cutoff filter. The as-synthesized catalyst powder (0.05 g) was dispersed in a 50 mL solution containing 0.2 M Na_2_S and 0.5 M Na_2_SO_3_, and 800 μL of potassium chloroplatinate was added as a cocatalyst. Before the reaction, the system was bubbled with nitrogen for 20 min to eliminate the air inside the system and ensure that the entire system was under anaerobic conditions. Using high purity nitrogen as the carrier gas, the photocatalytic H_2_ generation performance was determined by TCD gas chromatography (GC-7920) (Zhongjiao Jinyuan Technology Ltd, Beijing, China). Blank experiments were performed without a catalyst or light.

### 2.6. Theoretical Calculation

DFT calculations were performed in the Vienna ab initio simulation package (VASP). A spin-polarized GGA PBE functional [41], all-electron plane-wave basis sets with an energy cutoff of 520 eV, and a projector augmented wave (PAW) method were adopted [42,43]. A (3 × 3 × 1) Monkhorst–Pack mesh was used for the Brillouin-zone integrations to be sampled. The conjugate gradient algorithm was used in the optimization. The convergence threshold was set 1 × 10^−5^ eV in total energy and 0.02 eV/Å in force on each atom. In the simulations, the non-periodic boundary condition is employed, and the molecular model of a orthorhombic In_2_O_3_ (1 × 1 × 1) and hexagonal CdO (2 × 2 × 1) were established by using Materials Studio. The orthorhombic In_2_O_3_ and hexagonal CdO composite material were formed by using Materials Studio. To find the thermal stable morphology and achieve a conformation with minimum potential energy, energy minimization was performed, and these minimum energy conformations were used as the initial status in the following electronic structure simulations [44] and Materials Studio software was used for visualization and plotting. The adsorption energy change (ΔEabs) was determined as follows:ΔE_abs_ = E_total_ − E_slab_ − E_mol_
where Etotal is the total energy for the adsorption state, Eslab is the energy of pure surface, and Emol is the energy of adsorption molecule.

The free energy change (ΔG) for adsorptions was determined as follows:ΔG = E_total_ − E_slab_ − E_mol_ +ΔE_ZPE_ − TΔS
where Etotal is the total energy for the adsorption state, E_slab_ is the energy of pure surface, Emol is the energy of adsorption molecule, ΔE_ZPE_ is the zero-point energy change, and ΔS is the entropy change. As the vibrational entropy of H* in the adsorbed state is small, the entropy of adsorption of 1/2 H_2_ is S_H_ ≈ −0.5S_0H2_, where S_0H2_ is the entropy of H_2_ in the gas phase at the standard conditions. Therefore, the overall corrections were taken as in ΔG_H*_ = E_tota_l − E_slab_ − E_H2_/2 + 0.24 eV, where E_H2_ is the energy of H_2_ in the gas phase.

## 3. Results and Discussion

### 3.1. Structures and Morphologies Characterization of S/CdO@In*_2_*O*_3_* Nanofibers

The phase structures of the synthesized catalysts were studied using XRD, as shown in Figure 2a. The characteristic diffraction peaks of bare In_2_O_3_ at 21.5°, 30.6°, 35.5°, 51.0° and 60.7°ewere consistent with the standard card (JCPDS No. 71-2194) [45]. In contrast, the diffraction peaks of the CdO samples were located at 33.0°, 38.3°, 55.3°, 65.9° and 69.2°, and were indexed to the cubic phase (JCPDS No. 05-0640). The diffraction results show the peak of pure CdO is much sharper than that of individual In_2_O_3_, indicating that CdO has much better crystallinity. It can be observed that the prepared composites contain the characteristic diffraction peaks of both In_2_O_3_ and CdO species. This implies that the as-constructed hybrid catalyst contains metal oxides of In_2_O_3_ and CdO. After electrospinning with sulfourea, no variations were observed in the crystalline structure of CdO@In_2_O_3_, which may be attributed to its low S content. Infrared and Raman characterizations were carried out to study the chemical bonds and structural properties of the catalysts. Figure 2b represents the FTIR spectra of the samples. A strong band was observed at 3458 cm^−1^, corresponding to the O-H group stretching vibration of CdO. The strong and sharp IR peaks at 668 and 1638 cm^−1^ may be caused by C-O stretching vibration. For In_2_O_3_, the peaks at 447, 540, and 606 cm^−1^ correspond to the vibration of the In-O bond, whereas the peaks at 1638 cm^−1^ belong to the adsorbed H_2_O molecule. The peak at 1107 cm^−1^ can be attributed to the tensile vibration of the S species, indicating the successful doping of sulfur atoms into the CdO@In_2_O_3_ composite. The peak at about 1180 cm^−1^ should be attributed to the Cd-S bond vibration in the S/In_2_O_3_-CdO. Raman analysis (Figure 2c) reveals multiple characteristic peaks for In_2_O_3_ at 131, 305, 365, 494, and 627 cm^−1^. The peak at 131 cm^−1^ corresponds to the In-O vibration of the InO_6_ structural unit, and the peak at 305 cm^−1^ corresponds to the bending vibration of InO_6_ octahedron structural units. The peak at 365 cm^−1^ corresponds to the tensile vibration of In-O-In, and the peaks at 494 and 627 cm^−1^ correspond to the tensile vibration of the same octahedral InO_6_. Compared to the single materials, the peaks of CdO@In_2_O_3_ at 131 and 627 cm^−1^ appear to be blue-shifted, possibly because of the effect of oxygen vacancies on the vibration frequency. For CdO, the broad and strong peaks at 272 cm^−1^ are a combination of transverse phonons and optical phonons caused by the lattice perturbation of the CdO film. The peak observed at 569 cm^−1^, for the indium-doped CdO, may be related to the band crossing between the transverse optical mode and longitudinal optical mode of the vibration of the metal oxide (Cd-O) bond in the CdO film [46]. The peak observed at 949 cm^−1^ is also bound by the longitudinal optical band. According to the selection rule, it can be concluded that all characteristic peaks result from the second-order Raman scattering of the CdO species. The S/CdO@In_2_O_3_-25 composite exhibits a slight red shift at 272 cm^−1^ owing to lattice distortion. The pyrolysis of sulfur leads to increased hypoxia and lattice distortion in CdO-In_2_O_3_. The Raman spectra of the CdO@In_2_O_3_ composites are similar to that of CdO, possibly because of the low content of In_2_O_3_ or the coating of CdO on the In_2_O_3_ surface_._ The additional weak band at 602 cm^−1^ is caused by localized defects comprising oxygen vacancies in the hybrid. Sulfur doping induces more oxygen defects and lattice distortion in CdO@In_2_O_3_. These results indicate the successful preparation of S/CdO@In_2_O_3_ ternary heterojunctions.

Moreover, the morphology of the samples was analyzed using SEM and TEM. As shown in Figure 3a, the surface of the S/CdO@In_2_O_3_-25 precursor nanofibers is smooth with a relatively uniform diameter (about 660 nm). After calcination, the fiber surface became rough, and its diameter showed a downward trend with a porous structure owing to the decomposition of the raw materials (Appendix A). Furthermore, S/CdO@In_2_O_3_-25 maintained a fibrous morphology with nanosphere-attached surfaces, as shown in Figure 3b. Further TEM observations reveal that the morphology of S/CdO@In_2_O_3_-25 after calcination consists of a shorter fiber length and coarser surface (Figure 3c). Figure 3d shows that the lattice fringes of In_2_O_3_ and CdO are 0.295 and 0.271 nm, respectively [40]. The oxygen vacancies are shown in Figure 3e, where some red dotted circles represent the vacancy sites. Further elemental mapping analysis and EDS spectra suggest that the In, Cd, S, and O elements are evenly distributed on the entire S/CdO@In_2_O_3_-25 nanofibers, as shown in Figure 3f–i and Appendix A. The multiple components are closely combined, favoring the formation of ternary heterojunctions with close-contact interfaces.

Furthermore, the electronic interaction and elemental chemical states of the as-prepared samples were examined using XPS. As shown in Appendix A, In, Cd, O, S elements are found in the composites, as expected, and no other impurities are detected, which agrees well with the EDS analysis results. This indicates that S/CdO@In_2_O_3_-25 hybrid nanofibers were successfully synthesized by in situ electrospinning. From the In 3d spectrum (Figure 4a), it is found that the binding energies (BEs) of pure In_2_O_3_ are 444.01 and 451.56 eV, which correspond to In 3d_5/2_ and In 3d_3/2_, respectively [47]. After coupling with CdO, the BEs become 444.12 and 451.67 eV. These two peaks also exist in the fine S/CdO@In_2_O_3_-25 spectra, with binding energies of 444.19 and 451.71 eV. Both exhibit a positive shift relative to that of In_2_O_3_ alone. The positive shift for the In 3d species may be caused by the strong interaction among the S, In, and Cd elements, which ameliorates the interfacial charge-transfer behaviors. In Figure 4b, the Cd 3d spectrum of pure CdO exhibits two asymmetric peaks at 403.91 and 410.60 eV, which may belong to the spin-orbit splitting of Cd 3d_5/2_ and Cd 3d_3/2_, respectively [48], indicating that the chemical states of Cd in the nanocomposite are +2 [49]. After binding with In_2_O_3_, the BEs become 405.09 and 411.78 eV. The BEs in S/CdO@In_2_O_3_-25, in which positive shifts might give rise to lattice distortion, are 405.66 and 412.42 eV. Figure 4c shows that the O 1s of bare CdO has two asymmetric peaks at 528.34 and 531.49 eV, and the main oxygen peak at 531.49 eV confirms that the O^2-^ oxidation state exists in CdO [13]. The small oxygen peak at 528.34 eV is attributed to chemically adsorbed oxygen. Moreover, the O 1s spectrum of bare In_2_O_3_ also has two asymmetric peaks located at 529.58 and 531.46 eV. The lower-energy peak at 529.58 eV is attributed to the lattice oxygen of In_2_O_3_, and the higher-energy peak at 531.46 eV is caused by oxygen defects [50,51]. After the combination of CdO and In_2_O_3_, the two peaks for O 1s exhibit a positive shift and are located at 529.69 and 531.73 eV. When the samples are doped with sulfur, the positive shifts of the peaks increase and the two corresponding peaks are located at 530.53 and 532.83 eV. This indicates that S doping can increase oxygen vacancies, which may be conducive to the rapid separation of the photoinduced charge carriers. Furthermore, the S 2p spectrum (Figure 4d) can be divided into two peaks at 162.42 and 169.05 eV. These peaks are part of the spin orbits of S 2p_3/2_ and S 2p_1/2_ of S^2-^ [52,53], which have complementary oxygen atoms in the lattice of CdO and In_2_O_3_, thereby confirming the presence of oxygen vacancies. The BEs region between 166 and 172 eV can be ascribed to the oxidized sulfur species, and the peak located at 169.05 eV implies the formation of the expected metal (M)-O-S bond in CdO@In_2_O_3_ [53]. The above XPS results further confirm the successful synthesis of the ternary S/CdO@In_2_O_3_ hybrid heterojunction.

### 3.2. Photocatalytic Performance and Mechanism of Hydrogen Evolution Analysis

The photocatalytic performance of the as-prepared catalysts was evaluated under visible light illumination, whereas the corresponding control experiments were performed without light or the catalyst. For the control experiments, no products were formed, indicating that the light source and photocatalyst were important components for effective photocatalytic H_2_ evolution. Furthermore, it was found that the parental In_2_O_3_, CdO, and CdO@In_2_O_3_-25 did not produce H_2_ gas, as displayed in Figure 5a. The samples doped with sulfur, the S/In_2_O_3_ and S/CdO samples, exhibited photocatalytic H_2_ generation activities with H_2_ yield rates of 3.6 and 203.4 μmol g^−1^ h^−1^, respectively. As expected, the S/CdO@In_2_O_3_-25 catalyst exhibited the best hydrogen production performance with the highest H_2_ production rate of 4564.5 μmol g^−1^ h^−1^ and an ultrahigh H_2_ yield of 9129.1 μmol·g^−1^ after a 2 h reaction, as summarized in Figure 5a and Appendix A. This activity might be attributed to the rich surface oxygen vacancy defects caused by doping S into the hybrid, and forming a type-II heterojunction, which is conducive to the rapid separation/transfer of photo-induced charge carriers. As a result, the H_2_ evolution rate of S/CdO@In_2_O_3_ was higher than that of the S/CdO and S/In_2_O_3_ catalysts. Consequently, the as-synthesized hybrid heterojunction was essential for driving the efficient H_2_ evolution of S/CdO@In_2_O_3_. With an increase in the molar ratio of In_2_O_3_ to CdO, the H_2_ evolution rate of the S/CdO@In_2_O_3_ hybrid increased (Figure 5a), with a maximum value of 4564.5 μmol g^−1^ h^−1^ H_2_ obtained with S/CdO@In_2_O_3_-25. When the In_2_O_3_/CdO molar ratios exceeded 0.25, the H_2_ yield rate decreased, which might have been because the stacking of In_2_O_3_ accumulation influenced light absorption. This accumulation may lead to the collapse of the material structure. In addition, the H_2_ evolution efficiency of S/CdO@In_2_O_3_ with various molar ratios of In_2_O_3_/CdO is in good agreement with the PL emission peak strength. After four consecutive recycling reactions, the catalytic activity of S/CdO@In_2_O_3_-25 did not decrease and nearly 85% of the initial activity could be remained (Figure 5b). This strongly implies that the hybrid catalyst exhibited favorable photocatalytic stability, as further evidenced by the FTIR spectra (Appendix A). The relationship between the reaction time and generated H_2_ amount for the S/CdO@In_2_O_3_-25 was also determined, as shown in Appendix A. When the irradiation is 2 h, the hybrid had the best hydrogen evolution activity with the yield rate of 4564.5 μmol·g^−1^·h^−1^.

UV-Vis diffuse reflectance spectra were also analyzed to determine the origin of the photocatalytic activity of the sulphur-doped hybrid catalysts. As shown in Figure 6a, the absorption values of CdO and In_2_O_3_ are approximately 500 and 450 nm, respectively. These values are in good agreement with the values reported in previous studies [54,55]. Compared with the contradistinctive materials, the S/CdO@In_2_O_3_-25 composite exhibits a certain red shift. The improved visible light absorption properties may be related to the existence of oxygen vacancies and partial partially valence states. In other words, sulfur doping can effectively optimize the band gap because the hybridization of the O 2p and S 2p orbitals produces additional intermediate electronic states. The band gap was calculated using the Tauc curves, as shown in Figure 6b [56,57]. It can be concluded that the band gap energies (Eg) of S/CdO, S/In_2_O_3_ and S/CdO@In_2_O_3_-25 were 2.39, 2.54 and 2.22 eV, respectively. Further, photoluminescence (PL) spectroscopy was employed to study the separation of the photoexcited charge carriers. As depicted in Figure 6c, the steady-state PL spectra exhibit a wide peak at approximately 435 nm, resulting from the intrinsic energy band PL and surface oxygen defects. From the as-obtained PL results, it can be clearly seen that the original In_2_O_3_ exhibits the strongest PL intensity with a peak at 470 nm, owing to its significant charge recombination. For the CdO@In_2_O_3_ hybrid materials, the emission yield of In_2_O_3_ decreases significantly, which could be attributed to the strong interaction between In_2_O_3_ and CdO that contributed to the more efficient charge separation. By contrast, the S/CdO@In_2_O_3_-25 shows the lowest intensity, which strongly indicates that the introduction of S can effectively inhibit charge recombination, consistent with the previous photocatalytic performance. Furthermore, the transient photocurrent responses were determined to analyze the photoexcited charge carrier transfer properties of the materials. As shown in Figure 6d, the S/CdO@In_2_O_3_-25 catalyst possesses the highest photocurrent response compared to the other catalyst materials (Appendix A), which illustrates that the charge-separation properties in the S/CdO@In_2_O_3_-25 hybrid can be largely promoted. In addition, the S/CdO@In_2_O_3_-25 hybrid catalyst has the smallest semicircle compared to the other synthesized catalysts (Figure 6e), indicating that its charge transfer resistance is extremely low, which favors the separation and transfer of photo-induced charge carriers.

The absorption and desorption properties of the H* atom on the photocatalyst play a key role in the H_2_ evolution reaction. Density functional theory calculations were used to investigate this property in the as-prepared samples. As shown in Figure 6f, the calculated adsorption energies of the H* atom for S/CdO and S/In_2_O_3_ are 0.44 and −0.43 eV, respectively. In contrast, for S/CdO@In_2_O_3_-25, the adsorption energy of the H* atom is the smallest (−0.29 eV), which might be caused by the rich surface oxygen defects and abundant H* adsorbed on the S/CdO@In_2_O_3_-25 surface, as displayed in Figure 6g and Appendix A. The diminished adsorption energy of S/CdO@In_2_O_3_-25 is favorable for driving effective H_2_ evolution. Furthermore, the Mott–Schottky plot (Figure 7a–c) was applied to study the band structure properties of samples. The positive slopes of the tangent curves indicate that both of them were n-type semiconductors. Compared to the 0.21V vs. NHE (CdO) and −0.65 V vs. NHE (In_2_O_3_) (Appendix A), the flat band potentials of S/CdO and S/In_2_O_3_ reduce with −0.51 and −0.70 eV, respectively, which can be attributed to the introduction of sulphur with rich electrons. Thus, it can be inferred that the VB positions of S/CdO and S/In_2_O_3_ are 1.78 and 1.74 eV, respectively. According to the band gap energies tested in UV-DRS, the CB potential energies of the samples calculated by the formula Eg = E_VB_^−^ − E_CB_ are −0.61 and −0.80 eV, respectively. According to the same method, the Fermi levels, bandgap energy, and band structure of CdO and In_2_O_3_ are listed in Appendix A. Based on the above results, a type-II charge transfer mechanism is proposed (Figure 7c). Under simulated light irradiation, S/CdO and S/In_2_O_3_ produced excited electron-hole pairs. The generated electrons in S/In_2_O_3_ move to the CB of S/CdO, whereas the formed holes migrate to the VB of S/In_2_O_3_, leading to the accumulation of electrons in the CB of S/CdO and h^+^ in the VB of S/In_2_O_3_. In this case, photoreduction of H_2_O to H_2_ can be achieved using the appropriate redox potentials of S/CdO. Moreover, the adsorption or desorption behavior (|ΔGH*| → 0) of the S/CdO@In_2_O_3_ hybrid also facilitates the migration of photogenerated carriers, further promoting the separation of photoinduced electron-hole pairs.

## 4. Conclusions

Overall, a S/CdO@In_2_O_3_ hybrid with a ternary heterojunction was successfully synthesized for effective solar-driven H_2_ evolution. The experimental results illustrated that the S/CdO@In_2_O_3_-25 hybrid exhibited a remarkable H_2_ production rate of 4564.5 μmol g^−1^ h^−1^ under artificial sunlight irradiation, and this H_2_ yield rate was approximately 22.0 and 1261.0 times greater than those of the parental S/CdO and S/In_2_O_3_, respectively, surpassing that of many reported photocatalyst materials. Moreover, the as-designed S/CdO@In_2_O_3_ hybrid nanofibers and abundant oxygen defects could optimize the electronic structure and activation energies of the catalysts. Furthermore, a double charge-transfer mechanism based on the type-II charge-transfer mechanism was proposed to elucidate the prominent photocatalytic H_2_ generation activity. This study employed this in situ electrospinning approach to construct 1D S-doped CdO@In_2_O_3_ nanofibers for photocatalytic hydrogen evolution. These results of this study can be helpful in the development of efficient solar-to-fuel conversion materials.

## Figures and Tables

**Figure 1 nanomaterials-13-00401-f001:**
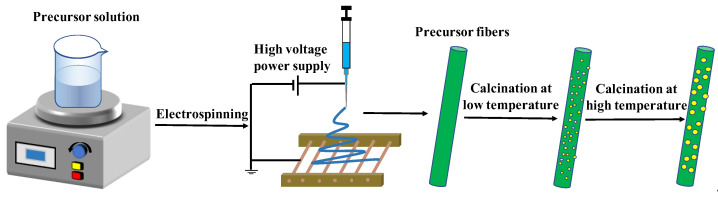
The scheme diagram of electrospinning process of S/CdO@In_2_O_3_ hybrid nanofibers during the calcinations process.

**Figure 2 nanomaterials-13-00401-f002:**
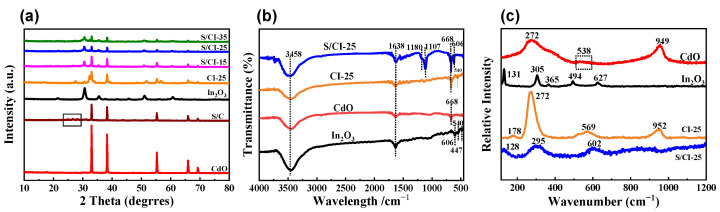
(**a**) XRD patterns, (**b**) FTIR spectra, and (**c**) Raman spectra of synthesized samples.

**Figure 3 nanomaterials-13-00401-f003:**
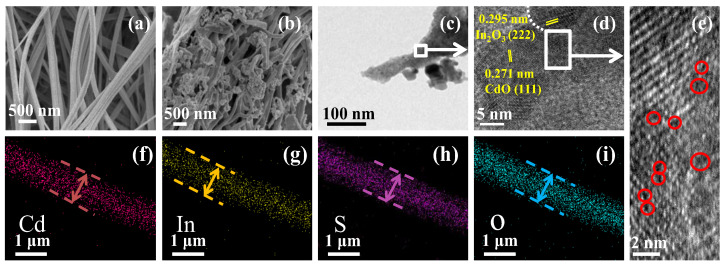
SEM images of S/CdO@In_2_O_3_-25 nanofibers (**a**) before calcination and (**b**) after calcination, (**c**–**e**) the corresponding TEM and HRTEM images and (**f**–**i**) EDS elemental mappings.

**Figure 4 nanomaterials-13-00401-f004:**
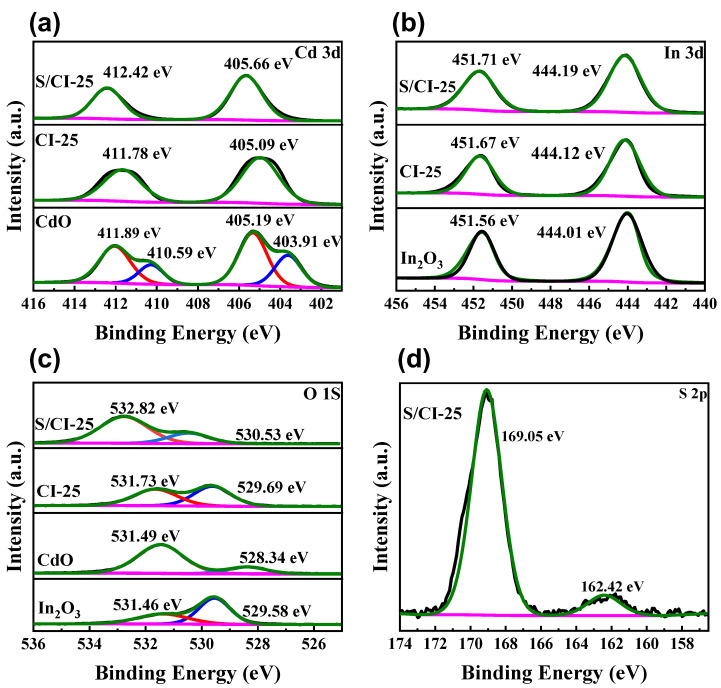
The XPS spectra of (**a**) Cd 3d, (**b**) In 3d, (**c**) O 1s, and (**d**) S 2p in the obtained samples.

**Figure 5 nanomaterials-13-00401-f005:**
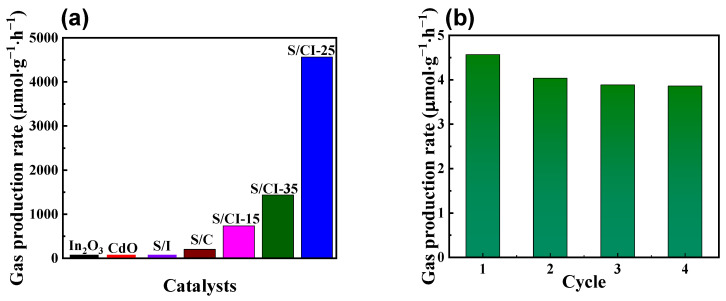
(**a**) H_2_ evolution rates of the as-synthesized photocatalysts and (**b**) recycling tests of photocatalytic H_2_ evolution rates over S/CdO@In_2_O_3_-25.

**Figure 6 nanomaterials-13-00401-f006:**
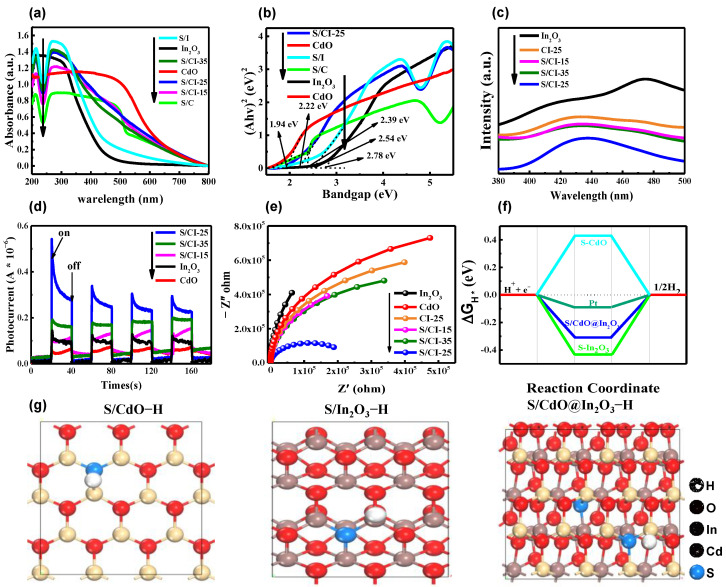
(**a**) UV/Vis diffuse reflectance spectra of samples, (**b**) the corresponding Tauc plots, (**c**) PL spectra, (**d**) transient photocurrent responses, (**e**) EIS spectra, (**f**) H_2_ adsorption energies determined by the DFT calculations of the as-prepared samples, and (**g**) their optimized H* atom adsorbed geometrical structures.

**Figure 7 nanomaterials-13-00401-f007:**
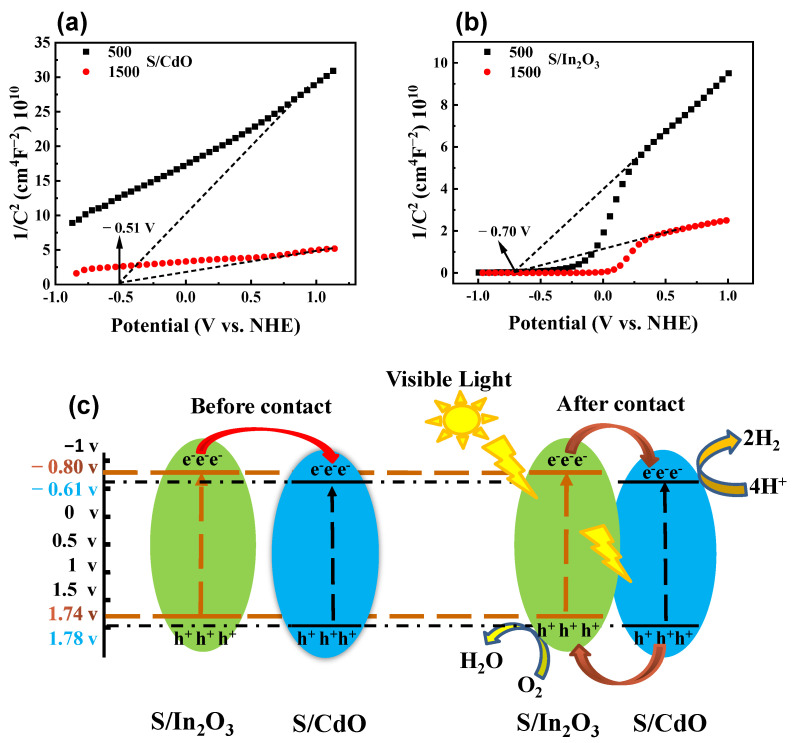
Mott–Schottky plots of (**a**) S/CdO, (**b**) S/In_2_O_3_, (**c**) XPS energy band structure, and possible charge transfer and photocatalytic mechanisms of the as-prepared S/CdO@In_2_O_3_-25 hybrid.

## Data Availability

The data supporting the findings of this study are available within the article and its Appendix A.

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
