# Peer review of "Constructing the Sulfur-Doped CdO@In2O3 Nanofibers Ternary Heterojunction for Efficient Photocatalytic Hydrogen Production"

_nanomaterials, 2023, doi:10.3390/nano13030401_

Round 1

Reviewer 1 Report

The present article represents an interesting and complete study on ternary heterojunction  based on S doped CdO/In2O3 for efficient photocatalytic hydrogen production. S-doped CdO@In2O3 nanofiber has been successfully prepared by the in-situ electro-spinning and the calcination process. Notably, the S/CdO@In2O3 displayed superior photocatalytic hydrogen evolution rate of 4564.58 μmol·g−1·h−1 under the artificial sunlight illumination, which was about 22.0 and 1261.0-folds higher than those of the comparative S/CdO and S/In2O3 samples, re-spectively. Based on detailed experimental studies and theoretical calculations, a detailed mechanism was proposed.

Therefore, I propose to accept the article after minor revisions.

1. What is the size of the CdO and In2O3 crystallites?

2. What is the influence of S doping on the crystallite size?

3. Please provide the XRD parameters.What is the influence of doping with S on the parameters of the non-doped components, of the doped ones and on the heterojunction?

4. Where are the S ions located inside the CdO and In2O3 lattice: interstitial, substitutional?

5.Please verify the Fig 3.d-e scale. It appears to be too large compared to the interplanar distance.

6. Concerning the FT-IR measurement, what it is the origin of the peak 1180 cm-1 marked on the fig?

7. The Eg value for S/CdO/In2O3 is 2.22 eV as mentioned in the text. In the figure these value in attributed to S/CdO/ In2O3 -25. Please make the correction in the text.                                                                     

8.Concerning the measurements of H2 production, an almost liniar increase of the H2 amount with irradiation time is observed? How the H2 amount evolves if the irradiation time is longer than 2h ( continues to increase or saturates?)

9. It would be useful to have a comparative presentation of the H2 generation efficiency with that of other CdO-based heterojunctions or In2O3- based heterojunctions reported in the literature

10.The conclusions are presented too briefly. I think it could be improved

Author Response

Please see the attached Response to Reviewer 1.docx file.  

Reviewer 2 Report

The scientific work is well done and the results are correct.

The English writing is poor and needs to be reviewed.

Author Response

Please see the attached Response to Reviewer 2.docx file.

Reviewer 3 Report

The results are interesting and give new insight into photocatalytic water-splitting on the heterojunction structure of S-doped CdO@In2O3. However, the reviewer strongly feels that the authors need to reconsider the accuracy of the measured values within the energy resolutions of the instruments (XPS, UV/Vis, EIS Flat-Band Potential for Mott-Schottky plots) to discuss the results more accurately. Furthermore, the oxygen-defect effect on the photocatalytic reaction does not clearly interpret the efficient charge-separation mechanism of the experimental results.   Please refer to the comments below.

1) Please carefully determine the bandgap energy values by the UV/Vis spectra, the valence band energies by the XPS, and the flat-band potentials, shown in Figs. 4, 6, and 7. The obtained values and significant figures seem to exceed the energy resolution of these instruments. The reviewer feels that the authors have ignored the significant experimental errors when they reported the measured values. They should refer to the instrumental resolution limits. For example, the measured XPS energy usually includes an error of 0.1 eV or 0.2 eV according to the electrical conductivity, sample surface condition, etc.

2) The authors did not show the stoichiometric production of oxygen to show photocatalytic water-splitting. Please discuss this point.

3) Can the production rate of hydrogen be six-digit accuracy? If so, the measured accuracy of the hydrogen production must be 2.2 x 10-4 cm3/(g h). Is this correct?

4) 2.4. Photoelectrochemical measurements: What is three-electrode quartz "battery"? How did the author prepare the catalyst films?  These must be clarified.

5) 2.5. Photocatalytic performance measurement: What is "H2 precipitation"?

6) Fig. 2a: "(  )" must be deleted because hkl here is the reflection index, not the crystal planes. Many papers show this mistake. Please follow the rule of X-ray crystallography.

7) Figs. 3(d) and (e): The images are unclear to insist that oxygen vacancies are formed. Can the authors show the heterojunction of In2O3 and CdO in the lattice image in Fig. 3(d)?  Microdiffraction patterns or EDX analysis seem to reveal this structure formation.

8) Fig. 4: Even for the high-resolution XPS, sample preparation for setting on a holder and the sample electrical conductivity can significantly affect the accuracy of measurement. Please carefully describe this in Section 2.

9)  Page 9 and Figs. 6(f),(g): There is no precise description of the DFT calculation.  The authors at least describe the conditions, e.g. cell size and the function they used, to discuss their results more accurately.  

Author Response

Please see the attached Response to Reviewer 3.docx file.

Round 2

Reviewer 3 Report

Since the manuscript has been improved, it can be accepted. However, the reviewer cannot still judge whether significant figures and rounding of the measured energy values are appropriate or not to support the conclusions. Therefore, we always need to pay careful attention to the accuracy of the measured values.